# Analysis of Hybrid MCDM Methods for the Performance Assessment and Ranking Public Transport Sector: A Case Study

**Swati Goyal** [1,*], **Shivi Agarwal** [1], **Narinderjit Singh Sawaran Singh** [2], **Trilok Mathur** [1] and **Nirbhay Mathur** [3]

1   Department of Mathematics, BITS Pilani, Pilani Campus, Rajasthan 333031, India
2   Faculty of Data Science and Information Technology, INTI International University, Persiaran Perdana BBN, Putra Nilai, Nilai 71800, Malaysia
3   Department of Electrical & Electronics, University Teknologi Petronas, Seri Iskandar 32610, Malaysia
*   Correspondence: swati.goyal@pilani.bits-pilani.ac.in

**Abstract:** The quality of the public transport sector affects the economy and the daily livelihoods of passengers. One of the most important objectives of policymakers is to choose the influencing criteria for performance evaluations. A variety of factors are crucial for raising the standards of public transportation services. In this investigation, we used a decision-based model with uncertainty in order to identify significant criteria in the public transport sector. We also performed a comparative analysis to rank the Rajasthan State Road Transport Corporation (RSRTC) bus depots based on their performance using hybrid multi-criteria decision-making (MCDM) techniques such as TOPSIS, VIKOR, and ELECTRE. To handle judgement ambiguities, in this work we incorporated the Delphi method (DM) and the analytic hierarchy process (AHP), along with fuzzy set theory. The fuzzy Delphi method was used to filter the important criteria. Using a fuzzy AHP approach, the screening criterion weights and rankings were determined. Furthermore, the bus depots were ranked using TOPSIS, VIKOR, and ELECTRE. Our findings can be applied in assisting policy-managers in formulating appropriate policies targeted at improving the overall health and competitiveness of bus depots using significant criteria and associated key indicators. In this study we investigated performance measures and proposed recommendations for the sustainable development of transportation in India.

**Keywords:** fuzzy set theory; multi-criteria decision making (MCDM) techniques; Delphi method (DM); analytic hierarchy process (AHP); public transport sector





## 1. Introduction

Public transportation is critical for advancing economic and social development since it improves mobility, which is widely acknowledged as one of the most basic and essential human needs [1]. It expedites the movement of passengers and freight from one location to another, thus contributing to the nation's economy [2]. Additionally, public transportation boosts production and growth by reducing transportation expenses, road and parking facility expenses, vehicle operating costs, accidents, and pollution. It also helps to meet social needs and provide transit services. This type of service is critical for connecting urban and rural areas, airports, train stations, and ports. Additionally, this sector employs operators of loading and unloading operations, roadside amenities, cleaners, conductors, and booking agents [3,4]. As a result, the World Bank dubbed public transit the "wheels of economic productivity" (World Bank 1994).

India is one of the nations with the greatest economic growth and has the fifth-largest economy in the world (https://bit.ly/39Wweqm, accessed on 10 July 2022). More than 300 million passengers/day travel through public transportation, providing livelihoods to about 40 million people. This sector alone contributes to almost 8% of GDP (http//www.financialexpress, accessed on 20 August 2020). Due to urbanization, this sector is facing formidable pressure to keep pace with increasing demand and unprecedented challenges in various aspects, including operation and service quality, availability, comfort, cost, and

safety. The deterioration of service levels may have an impact on the number of users, which consequently will also influence their contribution to GDP.

### 1.1. MCDM Techniques in the Public Transport Sector

One of the most crucial steps in the performance evaluation process is ranking and selecting the suitable parameters that affect the outcomes. MCDM offers several computational algorithms for the integrated ranking of alternatives. The performance of a service in the transport sector has been assessed frequently using MCDM methodologies. According to [5], 58 distinct MCDM approaches can be employed in private and public urban passenger transportation systems to make essential judgments in evaluating the design and operation of public transportation systems, which were made available between 1982 and 2014. In addition, the authors in [6] reviewed 89 papers studying transportation systems using MCDM techniques from 1993 to 2015. They found that it makes no difference which MCDM technique is better or worse, as the suitability of the methodology is determined by the circumstances of the specific decision.

Jamshidi et al. [7] assessed the efficiency and ranked the criteria that influenced passenger satisfaction through the use of the two-stage Delphi method in the road transport industry. A Delphi survey involving experts from academia, industry, and government revealed a diversified and multifaceted vision of future developments in Sweden for the year 2050 in regard to goods transport [8]. Diaz [9] improved the infrastructure and logistics service quality of road and port transport for the development of the export of goods using a micro-fixed-effects method for the period of 2012–2018 . Karam et al. [10] used a hybrid analytical method that combined meta-synthesis, an FDM, and AHP to improve the sustainability of the transportation industry. The performance of the metropolitan public transportation system was analyzed using a fuzzy multi-criteria analysis approach (MCAA) [11].

Relatedly, 276 publications' titles from the period of 1985–2012 were examined, and it was noted that 33% of research works used the AHP technique and developed a variation of it in the transportation sector. Yedle and Shrestha [12] employed AHP to assess six sustainable modes of transportation. Hawas et al. [13] employed the TOPSIS approach and the K-means clustering algorithm to develop strategies and reliable guidelines to improve public transit accessibility in urban cities. Ghorbanzadeh et al. [14] used Internal AHP to develop sustainable decisions in public transportation. A sensitivity analysis revealed that the factor ranking was very stable. Duleba and Moslem [15] investigated Pareto optimization using AHP to better analyze public preferences for supply quality in local bus transportation in Turkey. Moslem et al. [16] used fuzzy and interval AHP to gain a sustainable outcome for stakeholder groups in public urban transport development. The authors in [17] developed an integrated and sustainable framework by utilizing MCDM methods such as AHP and direct weighting for the urban bus system of Hyderabad. Moslem and Celikbilek [18] improved the service quality of public transport systems with the AHP and a multi-objective optimization method, implementing ratio analysis (MOORA) models with gray optimization. Kutlu et al. [19] proposed a new hybrid model based on picture fuzzy sets and linear assignment, while considering the level of indeterminacy of respondent evaluations in regard to decision alternatives. Streimikiene [20] proposed the interval TOPSIS method for the road transport sector. Aydin and Kahraman [21] focused on a hybrid of the fuzzy AHP and VIKOR MCDM techniques for the selection of public vehicles. Then, the use of AHP, fuzzy AHP, and ELECTRE was proposed by [22,23] for the evaluation of the selection of public transport. Avenali et al. [24] used a hybrid cost model that combined bottom-up and top-down methods to calculate unit standard costs for an Italian local public bus transportation industry. Güner [25] presented a two-stage AHP-TOPSIS technique for ranking alternatives on the basis of optimal situations and efficiency. Marchetti and Wanke [26] suggested a hybrid technique that combines TOPSIS with a genetic algorithm.

Subsidized fares, ineffective resource use, and a small number of under-performing bus depots all contribute to the current significant losses in the public transportation

sector. It would be fair to say that the transportation sector's financial performance is bleak and that a severe financial crisis is imminent. Therefore, it is imperative to evaluate the performance of each bus depot comprehensively and take the intensive steps needed to efficiently use the available resources and to provide better customer service and quality. Unfortunately, metrics to quantify the performance of bus depots are not available in the literature. Moreover, developing such a metric is challenging as various dimensions influence the performance of a bus depot, and hence it is often referred to as a multi-criteria problem. In many cases, performance evaluations for decision-makers involve uncertainty, with multiple and competing criteria.

Each approach has its own set of benefits and drawbacks, as well as its own region of application; no approach is superior to the others. The same multi-criteria decision problem may be solved by using more than one technique, resulting in more reliable decision data [27]. The aforementioned literature review highlighted the importance of MCDM techniques in performance assessments in the transport sector. Multi-criteria decision making (MCDM) methods have become one of the crucial decision-making techniques for dealing with real-world problems. The use of MCDM approaches to measure performance in the transportation sector is gaining attention worldwide. Hence, MCDM is an appropriate technique for dealing with complicated decision problems including many criteria, goals, or competition objectives. In fact, decision-makers always seek a criterion to determine the best decisions. Decision-making plays a vital role for managers at different organizational levels. However, developing suitable criteria for performance assessments depends on the statement of the problem.Further studies on proposing a metric to determine performance are limited.

In view of this, in this study we addressed the following questions:

**(a)** Which parameters could be the most effective for evaluating the performance of a bus depot?
**(b)** Which multi-criteria technique helps to identify the vital parameters and also to evaluate the significance of each parameter?
**(c)** How can we quantify the performance of a bus depot by aggregating all the attributes of the collected parameters?

Owing to the limitations in this field of study, in this investigation we aimed to advance a novel metric that can represent the performance of each bus depot. Through this study we aimed to provide a useful tool for policy-makers in creating policies for improving the overall health and competitiveness of RSRTC depots. A hybrid three-stage MCDM model, integrating the FDM, FAHP, and TOPSIS-VIKOR-ELECTRE approaches, was used to quantify the performance. Weights were assigned to the set of criteria after they had been established using FDM to reflect their relative relevance. The pair-wise comparison approach was used to compare the significance of two criteria. FAHP is a type of pair-wise comparison with a nine-pointscale for determining relative importance. Lastly, TOPSIS-VIKOR-ELECTRE was used to compare the ranking of depots using weights derived from FAHP. Detailed insights into the proposed technique are presented in the subsequent sections.

The detailed literature review on MCDM methods in the public transportation sector has been presented in Section 1.1. The study area, parameters, and fuzzy set theory are all explained in Section 2. Section 3 provides a comprehensive introduction to the fuzzy Delphi method, fuzzy AHP, and TOPSIS-VIKOR-ELECTRE methods, as well as the recommended methodology. Section 4 presents an evaluation of the comparative TOPSIS-VIKOR-ELECTRE results. Section 5 focuses on the weight selection process with a sensitivity analysis, and conclusions are drawn in Section 6.

## 2. Data Collection and Methodology Employed

### 2.1. Introduction of Study Area

In terms of geography, Rajasthan is the largest state in India. It is located between the northern latitudes of $23°30'–30°11'$ and the eastern longitudes of $69°29'–78°17'$ in the western

part of the country. The population of Rajasthan was around 82.4 million in 2021 (https: //bit.ly/2WKHOSf, accessed on 10 July 2022). There are a total of 33 districts, out of which 12 districts are in the Thar Desert. Furthermore, the economy of Rajasthan is heavily reliant on travel and tourism; therefore, it is imperative to have a good road network for effective passenger transportation. Bus services are widely used in Rajasthan to meet the huge and growing passenger transport demand. To enhance the convenience of public transport, the Rajasthan State Road Transport Corporation (RSRTC) was formed on 1 October, 1964, in accordance with the terms of the Road Transport Corporation Act, 1950.

RSRTC is a long-term supplier of intercity transportation with the purpose of providing low-cost, appropriate, punctual, and structured services to the traveling public in the state. There were 8 bus depots and 421 buses covering 45,000 kilometers, carrying 29,000 passengers per day in 1964. Currently, 4500 buses across 52 bus depots travel over 1.6 million kilometres each day, carrying over 0.9 million passengers (https://bit.ly/3t41I7P, accessed on 10 July 2022). The spatial distribution of the existing bus depots is shown in Figure 1.

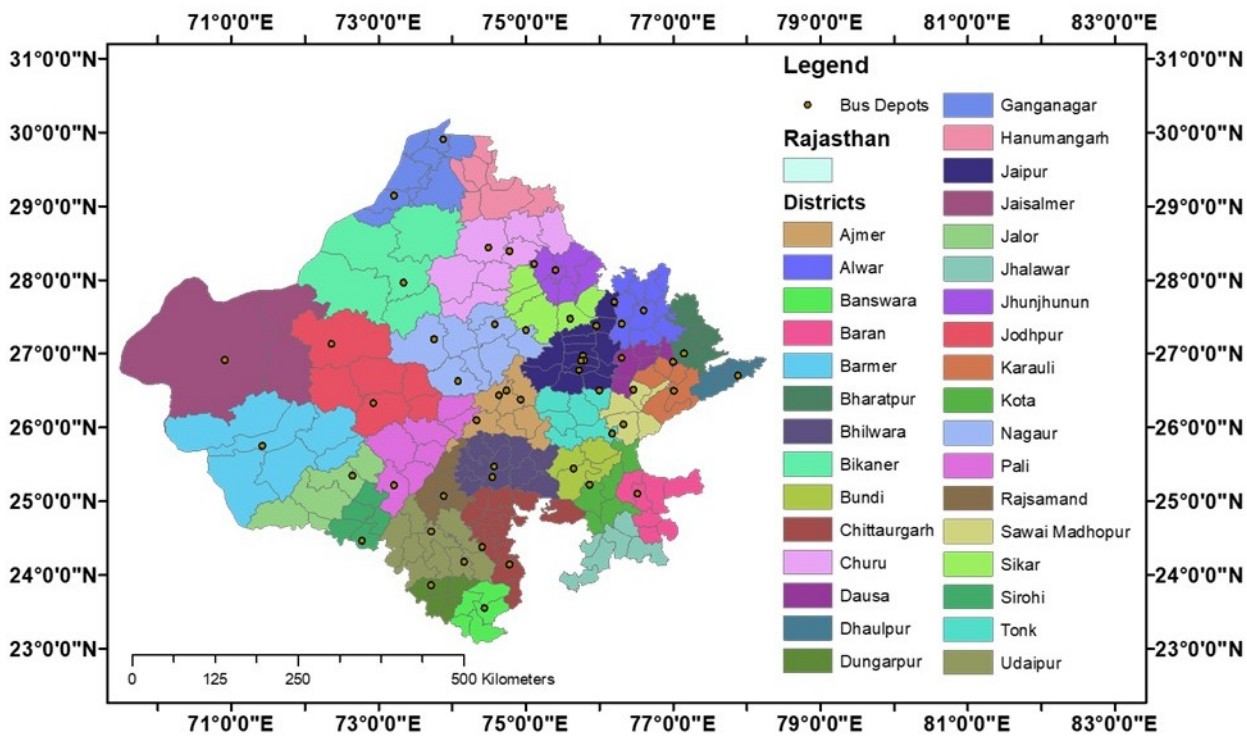

**Figure 1.** Spatial distribution of RSRTC bus depots.

### 2.2. Parameters

The choice of parameters is very important when conducting a performance evaluation. Considering the past research works, the critical parameters that influence the performance of a depot were determined. The aforementioned 29 parameters were classified into four key categories, namely, operational service, service quality, passenger service, and cost effects. In Table 1, every parameter is expressed with definitions, as well as a summary of the basic descriptive statistics (mean score, standard deviation, and variance).

### 2.3. Fuzzy Set Theory and Fuzzy Numbers

Ignoring uncertainty and impreciseness in data sets might diminish decision models' utility and predictive capacity. Goguen [28] proposed fuzzy set theory to indicate the linguistic term employed by the decision-maker and to reconcile the impreciseness, subjectivity, and unpredictability of human judgements. Fuzzy theory has been employed by decision-makers to deal with the uncertainties that develop during the data linguistics as-

sessment process. A fuzzy set is a "a class of objects" with a membership grade continuum, where the membership grade is an intermediate binary value between 0 and 1. Let $\tilde{A}$ be a fuzzy set of a universal set $X$ and such a set is distinguished with the membership grade function $\mu_{\tilde{A}}\{x\} : X \to [0, 1]$, which is associated with each element $(x = x_1, x_2, ..., x_n)$ in X, a real number lying between $[0, 1]$. The fuzzy set $\tilde{A}$ can be defined as:

$$A\tilde{(}x) = \{x, \mu_{\tilde{A}}\{x\}, \quad x \in X\} \tag{1}$$

**Table 1.** Definitions of parameters and summary of statistics of RSRTC bus depots for the year 2018–2019.

| Category | Criteria | Description | Mean | Std.Dev | Variance |
|---|---|---|---|---|---|
| Operational Service | Total vehicles | The number of vehicles held by a depot input. | 70.46 | 24.72 | 610.92 |
| | Scheduled vehicles | Total number of vehicles that were pre-assigned to a depot for that year | 80.41 | 27.1 | 734.33 |
| | Operated vehicles | Total number of vehicles that actually operated for a depot for that year | 55.75 | 21.02 | 441.88 |
| | Off-road vehicles | Total vehicles out of the number of operated vehicles that remained out of operation for a depot. | 10.25 | 6.13 | 37.6 |
| | Scheduled trips | Total count of trips scheduled for a depot for that year. | 80,402.75 | 33,344.19 | 1,111,834,768 |
| | Operating trips | Total trips actually operated in a year | 69,571.79 | 28,248.91 | 798,001,060.4 |
| | Extra trips | Unscheduled trips that operated in a year | 936.17 | 922.94 | 851,826.34 |
| | Curtailed Trips | Total count of cancelled trips | 13,632.75 | 8242.6 | 67,940,444.5 |
| | Total no. of employees | The number of employees in a depot, which is indicative of labour input. | 272.29 | 116.97 | 70.46 |
| | No. of routes | The number of routes, which is described as network size. | 42.89 | 14.08 | 198.34 |
| | Routes Distance | The route distance, which is described as total km travelled by passengers. | 9064.31 | 2966.59 | 8,800,646.88 |
| Passenger Service | Number of passengers | Total number of passengers who travelled in a year. | 59.38 | 28.03 | 785.42 |
| | Passenger km Occupied | The cumulative distance traveled by each passenger. | 3.9 | 1.5 | 2.26 |
| | Description of km | Total kilometers operated during a period, divided by the total number of buses in that particular period, and then divided by the number of days in the period. | 104.57 | 39.51 | 1561.01 |
| | Load factor | Percentage of total passenger kilometers in regard to the total carrying capacity. | 76.08 | 4.99 | 24.9 |
| Cost Effects | Income per seat per km (in lacs) | Total income divided by (average number of seats in a bus * km travelled). | 66.77 | 9.26 | 9.26 |
| | Total income per km | Total income divided by km travelled. | 3299.4 | 355.74 | 126,552.4 |
| | Operating income per km | Total operating income divided by km travelled. | 3254.65 | 349.2 | 121,941.96 |
| | Operating income (in lacs) | Operating income, also referred to as operating earnings. | 3447.68 | 1483.43 | 220,0561.01 |
| | Per vehicle per day income | income divided by total buses per day. | 12,824.6 | 2869.61 | 8,234,687.3 |
| | Total expenditure per km | Total expenditure divided by km travelled. | 3989.21 | 438.97 | 192,691.5 |
| | Profit/ loss per km | Total income per km-Total expenditure per km. | 689.77 | 400.46 | 160,367.04 |
| | | Diesel consumption km per liter. | 5.04 | 0.3 | 0.09 |
| | Consumption rate of diesel and oil | Engine oil top up km per liter. | 0.62 | 0.21 | 0.05 |
| | | Engine oil consumption per thousand km. | 12,824.6 | 2869.62 | 8,234,687.3 |
| Quality | Rate of breakdown | A measure of the mechanical reliability of a fleet, expressed in terms of the number of breakdowns per 10,000 kilometers. | 0.2 | 0.15 | 0.15 |
| | Rate of accident | The number of accidents per 100,000 kilometers. | 0.05 | 0.03 | 0 |
| | Punctuality | Percentage of scheduled trips that departed from the depot at their scheduled time. | 98.4 | 4.06 | 16.48 |
| | | Percentage of scheduled trips that arrived at the depot at their scheduled time. | 99.23 | 2.3 | 5.28 |
| | Fleet utilization | Percentage of buses on road in regard to the number of buses held by the depots. | 79.06 | 7.4 | 54.72 |
| | Vehicle utilization | Total kilometers traveled by a bus per day. | 391.87 | 46.39 | 2151.81 |
| | Tire efficiency | Ratio of km travelled to the maximum km possible for a tire | 91,860.48 | 22,437.66 | 503,448,490.2 |

In fuzzy inference systems, the fuzzy numbers that are considered in modelling the decision-making process, namely, triangular, trapezoidal, Gaussian, bell-shaped, sigmoid, Cauchy, and polynomial membership functions, are the most prominent. Because of its widespread acceptance in the literature, the triangular fuzzy membership number (TFN) was implanted, as it is themost notable and fundamental. They are simple to employ and ideal for promoting representation and information processing in a fuzzy environment.

The left and right sides of its linear representations are such that its membership function for a TFN $\tilde{A} = (a, b, c)$, $a < b < c$. The membership function is described below:

$$\mu_{\tilde{A}}(x) = \begin{cases} 0, & x \leq a, \ c \leq x \\ \dfrac{x-a}{b-a}, & a \leq x \leq b \\ \dfrac{c-x}{c-b}, & b \leq x \leq c \end{cases}$$

If $\tilde{A} = (a_1, b_1, c_1)$ and $\tilde{B} = (a_2, b_2, c_2)$ are two TFNs, then the basic mathematical procedures of these two TFNs are as follows:

$$(\tilde{A} + \tilde{B}) = (a_1 + a_2, b_1 + b_2, c_1 + c_2) \quad a_1, a_2 \geq 0 \tag{2}$$

$$(\tilde{A} - \tilde{B}) = (a_1 - c_2, b_1 - b_2, c_1 - a_2) \quad a_1, a_2 \geq 0 \tag{3}$$

$$(\tilde{A} \times \tilde{B}) = (a_1 \times a_2, b_1 \times b_2, c_1 \times c_2) \quad a_1, a_2 \geq 0 \tag{4}$$

$$(\tilde{A} \div \tilde{B}) = (a_1 \div a_2, b_1 \div b_2, c_1 \div c_2) \quad a_1, a_2 \geq 0 \tag{5}$$

$$(\tilde{A}^{-1}) = \left( \frac{1}{c_1}, \frac{1}{b_1}, \frac{1}{a_1} \right) \text{Inverse of a triangular fuzzy number.} \tag{6}$$

## 3. Research Methodology

In this study we present a sophisticated strategy which can be used for any of the decision-based transport sectors mentioned above. Figure 2 presents the steps involved in the proposed methodology.

A three-phase research methodology built on the theories of FDM, FAHP, TOPSIS-VIKOR-ELECTRE, and hybrid techniques was developed. In the fundamental stage of analysis, we carry out the identification of criteria related to the performance of the RSRTC bus depot. The list of criteria identified through the judgement of experts, aiming to capture their perception of each criterion, is given in Table 1. They were given the flexibility to mark their responses either by using a linguistic scale or a numeric scale, as mentioned in Table 2. Since representing perceptions using a crisp response is challenging, this study employed the concept of fuzzy numbers. Furthermore, the fuzzy Delphi method (FDM) was employed to analyze the obtained responses, and the significant criteria were identified in phase 1.

**Table 2.** Linguistic terms and corresponding TFNs for the significance weighting of criteria.

| Linguistics Term | Corresponding TFN |
|---|---|
| Very High Importance | (0.9, 1, 1) |
| High Importance | (0.7, 0.9, 1) |
| Medium High Important | (0.5, 0.7, 0.9) |
| Medium Importance | (0.3, 0.5, 0.7) |
| Medium Low Importance | (0.1, 0.3, 0.5) |
| Low Importance | (0, 0.1, 0.3) |
| Very Low Importance | (0, 0, 0.1) |

In phase 2, the significance of each relevant criteria was determined by implementing the FAHP method.

In the last phase, the criteria were used to determine the performance scores of the RSRTC depot and create a comparative ranking using a different three-stage hybrid MCDM technique (TOPSIS-VIKOR-ELECTRE).

### 3.1. Introduction to the Fuzzy Delphi Method

In the 1950s, the authors in [29] proposed the Delphi method at the Rand Corporation. The Delphi method is widely employed in management decision-making, prediction, analysis of public policy, and project organization to achieve the most accurate judgement among a group of experts. Furthermore, this method has been proven to be the most effective in detecting the trend of an enduring criterion. When investigating distributed group decisions, there is no apparent solution to a policy issue [30]. On the contrary, the Delphi approach allows for the full integration of multiple expert opinions. It is time-consuming and expensive, and a small group of experts cannot resolve all of the issues. It also has a lower rate of return because it attempts to obtain convergent answers through repeated surveys.

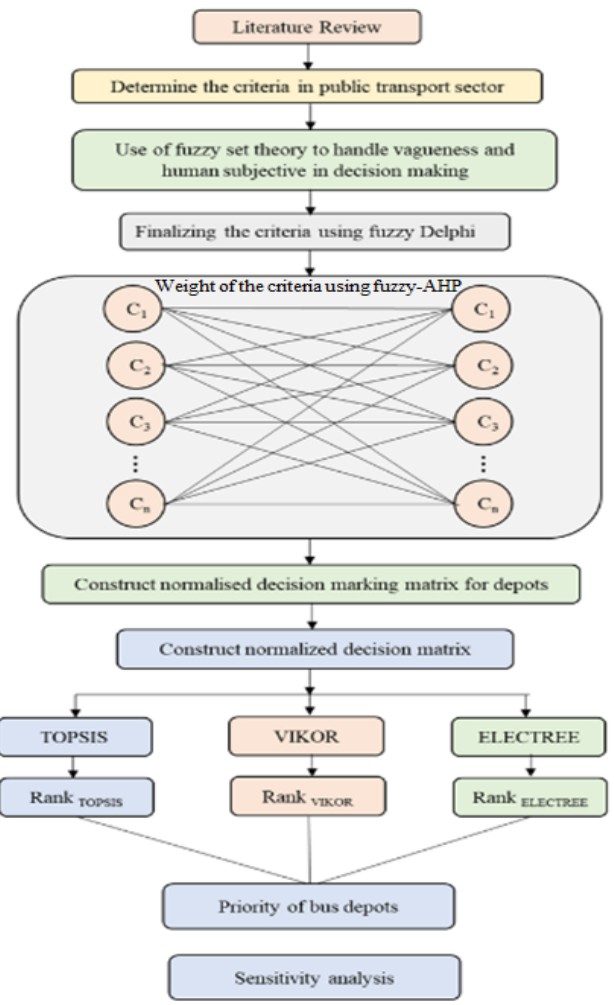

**Figure 2.** Schematic illustration of the proposed framework.

Murray et al. [31] developed the FDM by combining the Delphi technique and fuzzy set theory to improve upon the Delphi method's impreciseness and vagueness. Because it assigns the results in an objective manner and avoids different rounds of surveys, the FDM obtains the conclusion in a single round. To compute the statistically unbiased effect and minimize the effects of extreme values, the maximum and minimum values of expert opinions are used as the two terminal points of TFNs, and the geometric mean is taken as the TFN membership degree. FDM is entirely reliant on the opinions of a group of experts whose job it is to predict the outcome, which is usually accomplished by achieving a consensus without bringing the experts together face-to-face. In this study, we employed the FDM method to choose the most effective criteria, which was suggested by [32].

The details of the procedure of the FDM method used in this study can be described in five steps:

- **Step 1** Determine Criteria
  In this study we conducted a comprehensive literature review and used a conceptual framework to determine the criteria. Several literature-based parameters for performance measures in the transport sector were listed in a tabular format (Table 1).
- **Step 2** Collect Expert Judgements
  The expert judgement score for each criterion was considered on a linguistic scale (extreme importance, moderate importance, strong importance, and equal importance)

in a questionnaire. This questionnaire was divided into four sections: operational service, passenger service, cost effects, and quality.

- **Step 3** Establish Fuzzy Triangular Numbers

  In this study, we transformed linguistic assessments into TFNs. Linguistic criteria were chosen to analyze the relevance of each criterion based on Table 2. Assume that a fuzzy number represents the opinion of the $i$th expert of n experts $\tilde{Z}_{ij} = (a_{ij}, b_{ij}, c_{ij})$, $\forall\, i = 1, 2, ....., n, j = 1, 2, ......., m$, where m is the number of criteria. First, we compute the fuzzy weights of criteria $\tilde{A}_j = (a_j, b_j, c_j)$, $\forall\, j = 1, 2, ......., m$ as defined in the given equations,

$$a_j = \min_i a_{ij}$$

$$b_j = \left( \prod_{i=1}^{n} b_{ij} \right)^{1/n}$$

$$c_j = \max_i c_{ij}$$

  where indices $i$ and $j$ represent the number of experts and criterion, respectively.

- **Step 4** Defuzzification

  Defuzzification can be accomplished using a variety of complex approaches. The mean approach is one of the most simple and it is defined as in Equation (7),

$$M_j = \frac{(a_j + b_j + c_j)}{3}, \quad j = 1, 2, ......., m \tag{7}$$

  Hence, the defuzzified number $M_j$ quantifies the collective judgement of all experts based on the effectiveness of a criterion.

- **Step 5** Screening the Criteria

  Finally, by specifying a $(r)$ threshold, the appropriate criteria can be screened out of a large number of criteria. The screening criteria are given below:

  (a) If $M_j \geq r$, then add $j$th criterion into the evaluation index;
  (b) If $M_j < r$, then omit $j$th criterion from the list.

  The threshold of 0.6 was chosen for consideration as an evaluation criterion. The next round was selected if the total number of criteria was higher than or equal to 0.6; otherwise, it is discarded.

### 3.2. Fuzzy Analytic Hierarchy Process

AHP is an MCDM approach used to deal with complex systems, including decision-making among various choices, and it was first introduced by [33]. Professionals and academics have used AHP extensively in a variety of engineering and management applications. The primary object of the AHP technique is to break down an issue into smaller sub-problems on various levels and implement a hierarchical structure, moving down the levels from a goal to criteria, sub-criteria, and options. In the conventional AHP method, a nine-point scale is utilized to establish pairwise comparisons between criteria and sub-criteria. Nonetheless, due to the discrete scale, this technique is troublesome. For instance, AHP makes decisions based on specific data without taking into account the expertise and experience of an expert, and it enables the handling of uncertainty and vagueness [34].

To deal with the imprecision of expert assessments, FAHP adds fuzzy logic theory and the use of fuzzy numbers to the AHP technique.This strategy can assist in overcoming the drawbacks of the criterion weighting process. In the literature, various authors have presented several modifications of the FAHP approach and several applications. The first study that employed the fuzzy set theory in relation to AHP with fuzzy triangular numbers was suggested in [35]. Buckley [36] pioneered the application of trapezoidal fuzzy numbers to represent a decision maker's evaluation of alternatives for each criterion. Chang [37]

introduced a new approach to dealing with FAHP. In this study, we applied fuzzy triangular numbers to the pairwise comparison scale based on FAHP, as introduced by [37].

The FAHP approach consists of the following five steps:

- **Step 1** Hierarchy Structure
  The purpose of this step was to identify and rank the criteria that can lead to a shift away from private vehicles and toward public transit. We investigated three levels in the hierarchical structure, with the top-level containing the problem's aim. The middle layer contained the categories of criteria, whereas all of the public transport system's criteria, which were the results of the FDM approach, were contained in the bottom layer.

- **Step 2** Pairwise Comparison
  The interval consideration approach was used in this study to evaluate the range of ratings given by each expert. The fuzzy pairwise comparison matrix is used in linguistic responses, where experts decide the relative value of one criterion over another based on their expertise and experience. Researchers may use several approaches to aggregate expert judgments, such as the average method, the geometric mean method, the interval or range consideration technique, etc. The interval consideration approach was employed in this study to assess the range of rankings given by experts. TFNs were used to aggregate the expert rankings, with $i$ and $j$ describing the number of rows and columns, respectively, and $E$ describing the number of experts. Below are the expressions that were utilized to assess the ratings from the different experts.

$$\tilde{x}_{ij} = (l_{ij}, m_{ij}, n_{ij}) \quad \text{where} \quad i = 1, 2, \ldots, n, \quad j = 1, 2, \ldots, m, \quad e = 1, 2, \ldots, E \quad (8)$$

$$l_{ij} = \min_e l_{ije},$$

$$m_{ij} = \left( \prod_{i=1}^{n} m_{ije} \right)^{\frac{1}{n}}$$

$$n_{ij} = \max_e n_{ije}$$

- **Step 3** Fuzzy Weight Determination
  According to the extent, this analysiswas quantified through TFNs, as expressed in Equation (9), and was computed for an object set $X = x_1, x_2, \ldots, x_n$ and a goal set $G = G_1, G_2, \ldots, G_n$.

$$M_{G_i}^1, M_{G_i}^2, \ldots, M_{G_i}^m, \quad i = 1, 2, \ldots, n \quad (9)$$

The fuzzy synthetic extent value for the $i$th object was determined as follows in Equation (10)

$$S_i = \left( \sum_{j=1}^{m} l_j, \sum_{j=1}^{m} m_j, \sum_{j=1}^{m} n_j \right) \odot \left( \frac{1}{\sum_{i=1}^{n} n_i}, \frac{1}{\sum_{i=1}^{n} m_i}, \frac{1}{\sum_{i=1}^{n} l_i} \right) \quad (10)$$

- **Step 4** Degree of Possibility
  The degree of possibilities of $M_1 = (l_1, m_1, n_1) \leq M_2 = (l_2, m_2, n_2)$ was interpreted as

$$V(M_1 \leq M_2) = hgt(M_1 \cap M_2) \quad (11)$$

$$= \begin{cases} 1, & \text{if } m_1 \leq m_2 \\ \frac{n_1 - l_2}{(n_1 - m_1) + (m_2 - l_2)}, & \text{if } l_2 \geq n_1 \\ 0, & \text{otherwise} \end{cases} \quad (12)$$

To compare $M_1$ and $M_2$, we require both the values of $V(M_1 \leq M_2)$ and $V(M_2 \leq M_1)$. The degree of possibility $V(M_1 \geq M_2, M_3, \ldots, M_e)$ for a convex fuzzy number M and

$M_i$ $(i = 1, 2, ..., e)$ can be defined by:

$$V(M \geq M_1, M_2, ..., M_e) = V[(M \geq M_1) \text{ and } (M \geq M_2) \text{ and } ..... \text{ and } (M \geq M_e)]$$
$$= \min[V(M \geq M_i)], \quad i = 1, 2, 3, ..., e. \tag{13}$$

Now consider that,

$$d'^{(A_i)} = [\min[V(M_1 \geq M_e)], \min[V(M_2 \geq M_e)], ......., \min[V(M_n \geq M_e)]], i = 1, 2, ...., n; e \neq i \tag{14}$$

The weight vector for the object $(i = 1, 2, ...., n)$ can be calculated as follows:

$$W' = (d'^{(A_1)}, d'^{(A_2)}, ....., d'^{(A_n)})^T, \tag{15}$$

where the normalized weighted vector $W$ is a non-fuzzy number.

$$W = (d(A_1), d(A_2), ....., d(A_n))^T \tag{16}$$

- **Step 5** Consistency Ratio
  Priorities are meaningful only if they are derived from consistent matrices. Consistency indicates that pairwise comparisons are nearly as logical as random selections. The consistency index ($CI_k$) was calculated using the following equation, which originated from the most extensive eigenvalue method ($\lambda_{max}$) and which was introduced by [38]. The value of the consistency ratio ($CR_k$) should be less than 0.1 for consistent weights; otherwise, the corresponding weights should be re-evaluated to avoid inconsistency.

$$CI_k = \frac{\lambda_{max} - n}{n - 1} \tag{17}$$

$$CR_k = \frac{CI_k}{RI_k} \tag{18}$$

where the random index (RI) differs for each matrix size $n$. Table 3 was used to calculate the $n$ size consistency index matrix of a randomly generated pairwise comparison.

**Table 3.** Random consistency index.

| n | 1 | 2 | 3 | 4 | 5 | 6 | 7 | 8 | 9 | 10 |
|---|---|---|---|---|---|---|---|---|---|---|
| RI | 0 | 0 | 0.58 | 0.89 | 1.12 | 1.24 | 1.32 | 1.41 | 1.45 | 1.49 |

### 3.2.1. TOPSIS Method

Hwang and Yoon [39] offered a method for ranking the alternatives across several criteria, known as the technique for order of preference by resemblance to an ideal solution (TOPSIS). According to [40], the TOPSIS method is the second most famous MCDM method. The most desirable outcome must be the one that is furthest away from both the positive and the negative ideal solutions in order for the TOPSIS strategy to be successful [41]. In contrast to the negative ideal solution, which maximizes cost criteria at the expense of benefit criteria, the positive ideal solution maximizes benefit criteria while minimizing cost criteria. The distances from the ideal solutions, both positive and negative, are computed simultaneously. A preference ranking is created based on their comparative closeness and the sum of these two distance values.

This method was applied in this experiment in seven steps, which were as follows:

- **Step 1** Decision Matrix
  The decision matrix was determined.
- **Step 2** Vector Normalized Decision Matrix (VNDM)
  The decision matrix was "normalized" by translating different scales and units among different criteria into a common measurable unit to allow comparisons between the criteria.

- **Step 3** Weighted Normalized Decision Matrix (WNDM)
  In this step, the columns of the normalized matrix were multiplied by the associated weights $w_j \in [0, 1]$.
- **Step 4** Positive and Negative Ideal Solutions
  The best preferable option was the positive ideal solution (PIS) $[S^+]$, whereas the worst preferable alternative was the negative ideal solution (NIS) $[S^-]$.
- **Step 5** Euclidean Distance Measure
  Euclidean distance was computed on the basis of PIS $[S^+]$ to NIS $[S^-]$ for each component from the ideal $(V_j^+)$ and non-ideal alternatives $(V_j^-)$.
- **Step 6** Relative Closeness Coefficient
  The relative closeness coefficient $(\xi_i^*)$ was calculated to define the an ideal solution $S_i$.
- **Step 7** Priority Ranking
  The alternatives having a lower relative closeness coefficient $\xi_i^*$ were preferred.

### 3.2.2. VIKOR Method

The vlsekriterijuska optimizacija i komoromisno resenje (VIKOR) technique was created to address the complex issues with MCDM, including several attributes with divergent and incompatible criteria (non-commensurable units), and was introduced in [42] in 1998. As a planned tool, VIKOR's distinctive structure is employed when decision experts are unable to adequately communicate their preferences during the system design phase. This approach offers the decision-maker a compromise ranking of attributes based on the closest approach to the 'ideal'solution using the initial weights of a problem with competing criteria.

Any attribute that is added or removed could affect the results of the VIKOR ranking. For both the opponent and the majority, this tactic preserves a minimum of personal regret and a maximum of group usefulness. Few research studies have been conducted to address the numerous application domains of VIKOR. For a number of case studies, the revised VIKOR approach was suggested by [43]. Ilangkumaran and Kumanan [44] employed VIKOR to select the best maintenance approach for a textile spinning facility. Anojkumar et al. [45] recommended the use of the VIKOR approach to choose pipe materials for the sugar industry. In order to rate Taiwan's renewable energy sources (RES), ref. [46] implemented VIKOR. The steps below offer an explanation of the mathematical algorithm used in VIKOR computations:

- **Step1** Normalized Decision Matrix
  The goal of normalization is to standardize the matrix entry unit.
- **Step2** Ideal Solutions
  The positive ideal solution (PIS) $f_i^+$ and the negative ideal solution (NIS) $f_i^-$ values of all criteria are computed.
- **Step 3** The values of $D_j$ and $R_j$
  These are the utility $(D_j)$ and regret $(R_j)$ measures for each attribute, where $w_i$ is the weight of $i$th criteria.
- **Step 4** Compute the value of $Q_j$

$$Q_j = \frac{w(D_j - D^+)}{(D^- - D^+)} + \frac{(1 - w)(R_j - R^+)}{(R^- - R^+)} \tag{19}$$

$D^+ = \min_j D_j,\ D^- = \max_j D_j,\ R^+ = \min_j R_j,\ R^- = \max_j R_j$.

where the solutions obtained by $D^+$ and $R^+$ correspond to the maximum group of utility and the opponent's minimum individual loss, respectively, and $w = 0.5$ is supplied as a weight for the approach of the "majority of criteria". However, $w$ is capable of setting any value between 0 and 1.

- **Step 5** Calculate the rank of the alternatives by means of the given ranking index $(Q_j)$ in decreasing order.

### 3.2.3. ELECTRE-I Method

The unique AHP-based optimal design approach, known as ELECTRE-I, was proposed in [47]. That study showed that AHP-based ELECTRE-I models may react effectively when competing criteria are present, and they are particularly useful for making decisions that call for widespread agreement. This approach to representing a decision maker's preferences across a variety of areas is known as ELECTRE-I. In addition to ELECTRE-I, several alternative approaches, such as ELECTRE-IV, ELECTRE-IS, ELECTRE-TRI, ELECTRE TRI-C, and ELECTRE TRI-N, have emerged from ELECTRE. Bojkovic et al. [48] used ELECTRE-I to examine the performance of transportation systems in relation to sustainable development challenges. To reduce the subjectivity of the decision-maker, they offered a variation of the ELECTRE approach. Veeramachaneni and Kandikonda [49] used the ELECTRE approach to compare two different public bus networks, one run by the local government and the other by private businesses. However, ELECTRE-I is unable to calculate the ranking of attributes. Electre-II was proposed to address this flaw in ELECTRE-I and create a ranking of alternatives. This study's methodology was divided into the following eight steps:

- **Step 1** Normalized Decision Matrix
  The normalization of the assessment matrix is the process of converting various scales and units across several criteria into common measurable units to enable comparisons across the criteria. To achieve this, a variety of normalized processes can be employed to construct an element $r_{ij}$ of the normalizing evaluation matrix R if $f_{ij}$ is the evaluation matrix R of alternative $j$ under the evaluation criterion $i$.

$$r_{ij} = \frac{a_{ij}}{\sqrt{\sum_{n=1}^{m} a_{ij}^2}} \quad i = 1, 2, \dots n; j = 1, 2, \dots, m \tag{20}$$

- **Step 2** Weighted Normalized Decision Matrix
  To produce the weighted normalized decision matrix, multiply the normalized evaluation matrix $r_{ij}$ with its associated weight $w_i$.

$$v_{ij} = w_i * r_{ij} \quad i = 1, 2, 3, \dots, n, \quad j = 1, 2, 3, \dots, m. \tag{21}$$

where $\sum_{i}^{n} w_i = 1^n = 1$

- **Step 3** Ascertainment of Concordance ($C_{pq}$) and Discordance ($D_{pq}$) Sets
  Let $A_i = \{p, q, r, \dots\}$ indicate a finite set of attributes. In the following formulation, the attribute sets are divided into two different sets: ($C_{pq}$) and ($D_{pq}$). If the following criteria are satisfied, the concordance set is used to describe the dominance query; after complementing $C_{pq}$, we obtain a set of discordance intervals ($D_{pq}$):

$$C_{pq} = \{j | a_{pj} \geq a_{qj}\}, \quad D_{pq} = \{j | a_{pj} \leq a_{qj}\} = \{j - C_{pq}\} \tag{22}$$

- **Step 4** Concordance Set Matrix
  The concordance interval matrix ($C_{pq}$) between $A_p$ and $A_q$ can be estimated based on the decision maker's preference for attributes. The concordance index is establised by means of the equation

$$C_{pq} = \sum_{j=C_{pq}} W_j \tag{23}$$

- **Step 5** Discordance Interval Matrix
  The discordance index ($D_{pq}$) can be interpreted as the existence of discontent in the choice of scheme '$p$' as opposed to '$q$'. In more detail, we can define

$$D_{pq} = \frac{\max\limits_{j \in D_{pq}} \mid V_{pj} - V_{qj} \mid}{\max\limits_{j \in m,n} \mid V_{mj} - V_{nj} \mid} \tag{24}$$

where $D_{pq}$ represents the discordance index and $m$, $n$ is used to compute the weighted normalized value among all target attributes.

- **Step 6** Concordance Interval Matrix
  The equation below expresses the concordance index matrix for satisfaction measurement:

$$\bar{c} = \sum_{p=1}^{m} \sum_{q=1}^{m} \frac{c(p,q)}{m(m-1)} \tag{25}$$

Hence, $\bar{c}$ is the critical value which is evaluated by means of the average dominance index. Thus, the Boolean matrix (F) is

$$F = \begin{cases} f(p,q) = 1 & \text{if } c(p,q) \geq \bar{c} \\ f(p,q) = 0 & \text{if } c(p,q) < \bar{c} \end{cases} \tag{26}$$

- **Step 7** Discordance Interval Matrix

$$\bar{d} = \sum_{p=1}^{m} \sum_{q=1}^{m} \frac{d(p,q)}{m(m-1)} \tag{27}$$

Based on the discordance index mentioned above, the discordance index matrix (E) is given by

$$E = \begin{cases} e(p,q) = 1 & \text{if } d(p,q) \leq \bar{d} \\ e(p,q) = 0 & \text{if } d(p,q) > \bar{d} \end{cases} \tag{28}$$

- **Step 8** Net Superior and Inferior Values
  The net superior ($\bar{c}_p$) adds together the numbers of competitive superiority for all attributes.

$$c_p = \sum_{q=1}^{n} C(p,q) - \sum_{q=1}^{n} C(q,p) \tag{29}$$

On the contrary, the inferior values ($\bar{d}_p$) are used to determine the number of inferiority for the ranking of the attributes.

$$d_p = \sum_{q=1}^{n} D(p,q) - \sum_{q=1}^{n} D(q,p) \tag{30}$$

## 4. Case Study: Performance of the Public Transport Sector

The results of each phase in this study, which took into account transportation system management, are displayed in Figure 2.

In the current study, the literature review, expert opinions, and the available data revealed 29 criteria connected to the adoption of performance measurement in the public transport sector. These can be grouped into the categories of operational service, quality service, passenger service, and cost effects. The FDM technique was used to deal with ambiguity in the finalization of the significant criteria. A questionnaire was prepared and developed to gather expert opinions. The experts chosen for our analysis had five years of experience in their respective fields. Four experts from academia were included because of their strong influence on policy decision-making. Finally, the results of all expert questionnaires were combined to form the overall judgments. The experts were asked to rate the influences of criteria on performance on a linguistic scale from 0 to 1, with

1 indicating that criteria had a significant impact on performance and 0 indicating that they had a low impact. The judgment of experts was captured using a scale shown in Table 2.

After defuzzification and filtering, we obtained crisp numbers that reflected the aggregate judgments of the experts, as shown in Table 4. A threshold value of r = 0.60 was used based on prior studies and expert consultation to determine whether a given criterion should be included or excluded . The criteria having a threshold value <0.60 were accepted (A); otherwise, they were rejected from the list. Fourteen vital criteria were identified, whereas 15 criteria were not accepted.

**Table 4.** List of accepted criteria based on FDM analysis.

| Category | Criteria | Average Fuzzy Weights $(a_j, b_j, c_j)$ | Defuzzification $(M_j)$ |
|---|---|---|---|
| C1: Operational Service | C11: Operated vehicle | (0.7, 0.95, 1) | 0.883 |
| | C12: Operating trips | (0, 0, 0.3) | 0.1 |
| | C13: Total no. of employees | (0.7, 0.97, 1) | 0.891 |
| | C14: Route Distances | (0.5, 0.81, 1) | 0.772 |
| C2: Service Quality | C21: Punctuality | (0.5, 0.87, 1) | 0.789 |
| | C22: Fleet utilization | (0.5, 0.89, 1) | 0.797 |
| | C23: Vehicle utilization | (0.7, 0.97, 1) | 0.891 |
| | C24: Rate of breakdown | (0.3, 0.69, 1) | 0.662 |
| C3: Passenger Service | C31: Number of passengers | (0.7, 0.95, 1) | 0.883 |
| | C32: Passenger km Occupied | (0.7, 0.95, 1) | 0.883 |
| | C33: Load factor | (0.7, 0.95, 1) | 0.883 |
| C4: Cost Effects | C41: Total income per k.m. | (0.3, 0.63, 1) | 0.643 |
| | C42: Operating income per km | (0.3, 0.73, 1) | 0.677 |
| | C43: Total expenditure per km | (0.5, 0.87, 1) | 0.789 |

### 4.1. Phase 1: Identification and Classification of Criteria Using the FDM Technique

As a result, the experts collectively ranked "Total no. of employees" and "Vehicle utilization" as the most vital criteria for operational service and service quality. Other criteria, "Income per seat per km (in lacs)", "Scheduled vehicles", and "No. of routes", failed to reach the threshold level. "Curtailed trips" and "Profit/ loss per km" were identified as the least relevant criteria. The process in this study excluded some of the significant criteria in judging the quality of public transportation services, such as the rate of accidents, total vehicles, scheduled trips, and kilometers traveled. These results imply that quality parameter preferences and rankings can vary in different regional settings and throughout time due to the expansion of transportation services and the relative influence of new modes.

### 4.2. Phase 2: FAHP Computations for Priority Weights

In this study, we listed the most significant criteria and suggested a method for evaluating the bus depot's performance. The decision hierarchy structure for the process of choosing the criteria had three levels.

The objective of the decision-making process was to determine the best bus depots at the first level of the hierarchy. At the second level, all criteria were divided into four categories, and at the third level, the significant weights of criteria were obtained. Using the filled-out questionnaire forms that were obtained from the four expert panel members, fuzzy AHP was used to determine the relative weights of the main categories and their criteria with respect to the aim. The FAHP methodology required expert relies on Saaty's linguistic nine-point scale and the evaluation of a pair-wise comparison matrix for each category. In the pair-wise comparison method, each criterion is compared to all other criteria using Equation (8).

The fuzzy comparison judgments of all categories and criteria in connection to the ultimate objective are displayed in Table 5. This was calculated using the geometric mean of the evaluation findings. Finally, a pair-wise comparison matrix was generated and a decision was made. The weights of the criteria were generated from this final comparison matrix. The consistency index and consistency ratio were calculated to determine whether the prioritization of the criteria in the pairwise comparison matrix was accurate. A decision-making group validated the weights at the end of this stage. In Table 6 the relative weights, consistency index, and consistency ratio of the criterion were derived.

**Table 5.** Fuzzy pairwise comparison matrix corresponding to the category.

| $\lambda_{max} = 4.18$, CR = 0.069 | | | | | |
|---|---|---|---|---|---|
| **Operational Service** | **Service Quality** | **Passenger Service** | **Cost Effects** | | **Local Weight** |
| Operational Service | (1.00,1.00,1.00) | (2.00,2.00,2.00) | (2.00,2.91,4.00) | (2.00,3.87,7.00) | 0.501 |
| Service Quality | (0.25,0.34,0.50) | (0.50,0.59,1) | (1.00,1.00,1.00) | (0.33,1.00,3.00) | 0.143 |
| Passenger Service | (0.50,0.50,0.50) | (1.00,1.00,1.00) | (1.00,1.68,2.00) | (2.00,2.21,3.00) | 0.242 |
| Cost Effects | (0.14,0.26,0.50) | (0.33,0.45,0.50) | (0.33,1.00,3.00) | (1.00,1.00,1.00) | 0.113 |

**Fuzzy pairwise comparison matrix and relative local weights corresponding to operational service**

| $\lambda_{max} = 4.18$, CR = 0.093 | | | | | |
|---|---|---|---|---|---|
| | Operated vehicle | Operating trips | Total no. of employees | Route Distances | Local Weight |
| Operated vehicle | (1.00,1.00,1.00) | (0.50,1.73,3.00) | (2.00,2.91,9.00) | (0.33,1.41,3.00) | 0.361 |
| Operating trips | (0.14,0.28,2.00) | (1.00,1.00,1.00) | (2.00,2.21,3.00) | (0.25,1.46,3.00) | 0.303 |
| Total no. of employees | (0.11,0.28,0.50) | (0.33,0.37,0.50) | (1.00,1.00,1.00) | (0.33,0.33,0.33) | 0.078 |
| Route Distances | (0.14,0.29,2.00) | (0.14,0.27,1.00) | (2.00,2.21,3.00) | (1.00,1.00,1.00) | 0.258 |

**Fuzzy pairwise comparison matrix and relative local weights corresponding to service quality**

| $\lambda_{max} = 4.18$, CR = 0.083 | | | | | |
|---|---|---|---|---|---|
| Punctuality | Vehicle utilization | Fleet utilization | Rate of breakdown | | Local Weight |
| Punctuality | (1.00,1.00,1.00) | (2.00,3.31,5.00) | (2.00,2.21,3.00) | (0.50,0.71,2.00) | 0.367 |
| Vehicle utilization | (0.20,0.30,0.50) | (1.00,1.00,1.00) | (0.50,1.19,2.00) | (0.33,0.45,0.50) | 0.119 |
| Fleet utilization | (0.33,0.45,0.50) | (0.50,0.84,2.00) | (1.00,1.00,1.00) | (0.33,0.58,2.00) | 0.179 |
| Rate of breakdown | (0.50,1.41,2.00) | (2.00,2.21,3.00) | (0.50,1.73,3.00) | (1.00,1.00,1.00) | 0.335 |

**Fuzzy pairwise comparison matrix and relative local weights corresponding to passenger service**

| $\lambda_{max} = 4.18$, CR = 0.082 | | | | |
|---|---|---|---|---|
| Passenger km Occupied | Number of passengers | Load factor | | Local Weight |
| Passenger km Occupied | (1.00,1.00,1.00) | (0.50,0.84,1.00) | (1.00,1.19,2.00) | 0.337 |
| Number of passengers | (1.00,1.19,2.00) | (1.00,1.00,2.00) | (1.00,1.68,2.00) | 0.444 |
| Load factor | (0.50,0.84,1.00) | (0.50,0.59,1.00) | (1.00,1.00,1.00) | 0.219 |

**Fuzzy pairwise comparison matrix and relative local weights corresponding to cost effects**

| $\lambda_{max} = 4.18$, CR = 0.092 | | | | |
|---|---|---|---|---|
| Total income per km | Total expenditure per km | Operating income per km | | Local Weight |
| Total income per km | (1.00,1.00,1.00) | (0.33,0.76,1.00) | (0.50,0.84,1.00) | 0.244 |
| Total expenditure per km | (1.00,1.32,3.00) | (1.00,1.00,1.00) | (1.00,1.00,1.00) | 0.388 |
| Operating income per km | (1.00,1.19,2.00) | (1.00,1.00,1.00) | (1.00,1.00,1.00) | 0.368 |

The values of $RI_k$, shown in Table 3, for three and four criteria were 0.58 and 0.89, respectively. The results presented in Table 6 show that the "Operational Service" (C1) category had the highest weight among the categories. The "Passenger Service" (C3) category was ranked as the second most important category. "Service Quality" and "Cost Effects" were ranked in the third and fourth place, respectively.

- **Operational Service (C1)**
  'Operated vehicle'(C11) had the highest priority, followed by 'Operating trips'(C12), 'Route Distances'(C14), and 'Total no. of employees'(C13).
- **Service Quality (C2)**
  'Punctuality'(C21) had the highest priority, followed by 'Rate of breakdown'(C24), 'Fleet utilization'(C23), and 'Vehicle utilization'(C22).

**Table 6.** Results of the calculation of weights by means of FAHP.

| Criteria | Local Weight Using FAHP | Sub-Criteria | Local Weight Using FAHP | Global Weight Using FAHP | Rank |
|---|---|---|---|---|---|
| Operational Service | 0.501 | Operated vehicle | 0.361 | **0.181** | 1 |
| | | Operating trips | 0.303 | **0.152** | 2 |
| | | Total no. of employees | 0.078 | **0.039** | 10 |
| | | Route Distances | 0.258 | **0.129** | 3 |
| Service Quality | 0.143 | Punctuality | 0.367 | **0.053** | 6 |
| | | Vehicle utilization | 0.119 | **0.017** | 13 |
| | | Fleet utilization | 0.179 | **0.026** | 12 |
| | | Rate of breakdown | 0.335 | **0.048** | 7 |
| Passenger Service | 0.242 | Passenger km Occupied | 0.337 | **0.082** | 5 |
| | | Number of passengers | 0.444 | **0.107** | 4 |
| | | Load factor | 0.219 | **0.053** | 6 |
| Cost Effects | 0.113 | Total income per km | 0.244 | **0.028** | 11 |
| | | Total expenditure per km | 0.388 | **0.044** | 8 |
| | | Operating income per km | 0.368 | **0.042** | 9 |

- **Passenger Service (C3)**
  'Number of passengers'(C32) had the highest priority, followed by 'Passenger km occupied'(C31), and 'Load factor'(C33)
- **Cost Effects (C4)**
  'Total expenditure per km'(C42) had the highest priority, followed by 'Operating income per km'(C43), and 'Total income per km'(C41).

*4.3. Phase 3: Overall Comparison of the Deterministic MCDM Methods for Performance Evaluation*

Based on the FAHP technique, the weights of all the criteria and rankings were determined. To rank all the depots, we applied the TOPSIS-VIKOR-ELECTRE methods. These were the three MCDM methods considered in this paper. Table 7 presents the rankings of the depots based on the final score values, as derived using the three MCDM algorithms.

- **TOPSIS Results**
  The results obtained by TOPSIS are tabulated in Table 7. Sikar was the best-performing depot, with highest performance score of 0.78, whereas Karauli is the worst-performing depot with smallest performance score of 0.02.
- **VIKOR Results**
  Table 7 illustrates the assigned rank for the depots based on the VIKOR index value and the outcomes. Similarly to the results shown above, in VIKOR, Sikar was the best-performing depot, with a performance value of 0.0465, whereas Karauli is the worst-performing depot with a performance value of 0.995. As an illustration, Sikar was placed in the top spot with aggregate depots and the value of the index was 0.9535 (1–0.0465), which was the closest value to the ideal solution of 1.
- **ELECTRE Results**
  The ranking of the RSRTC bus depots was determined using the inferior and superior values of ELECTRE. The obtained rankings are tabulated in Table 7. In ranking results obtained using ELECTRE, Alwar was the best performing depot (33.12), whereas Jaisalmer was the worst performing depot (−40.74) out of the 52 depots.

A decision matrix served as the beginning point for evaluating the rank of the depots. Using the MCDM technique, the efficient depots were evaluated in this study, taking into account a variety of competing criteria. The results showed that Sikar was the best performing depot, whereas Jaisalmer and Karauli were the worst performing depots according to all methods. Abu Road and Dungarpur had the same rankings in all three methods. These three methods were ranked in a comparable order, though not identically. TOPSIS and

VIKOR produced 97.6% similar rankings of the depots. VIKOR and ELECTRE produced 95.74% similar rankings of the depots.

**Table 7.** Scores and final rankings of bus depots obtained using each method.

| Methods | TOPSIS | | VIKOR | | ELECTRE | | |
|---|---|---|---|---|---|---|---|
| Bus Depots | Score | Rank | Score | Rank | Score | Rank | Final Rank |
| Abu Road | 0.09 | 46 | 0.68 | 46 | −25.98 | 46 | 47 |
| Ajaymeru | 0.21 | 7 | 0.21 | 6 | 17.62 | 11 | 8 |
| Ajmer | 0.6 | 5 | 0.19 | 5 | 21.78 | 6 | 5 |
| Alwar | 0.31 | 3 | 0.08 | 2 | 33.12 | 1 | 2 |
| Anoopgarh | 0.08 | 37 | 0.58 | 38 | −6.08 | 35 | 37 |
| Banswara | 0.1 | 35 | 0.48 | 29 | −2.88 | 28 | 33 |
| Baran | 0.15 | 26 | 0.48 | 28 | −4.08 | 33 | 28 |
| Barmer | 0.14 | 33 | 0.58 | 39 | −7.66 | 37 | 35 |
| Beawar | 0.22 | 25 | 0.56 | 36 | 3.75 | 24 | 27 |
| Bharatpur | 0.18 | 16 | 0.33 | 17 | 18.7 | 8 | 13 |
| Bhilwara | 0.13 | 17 | 0.3 | 13 | 10.65 | 14 | 16 |
| Bikaner | 0.13 | 10 | 0.28 | 12 | 10.15 | 16 | 12 |
| Bundi | 0.15 | 31 | 0.45 | 26 | −1.45 | 27 | 26 |
| Chittorgarh | 0.17 | 13 | 0.22 | 7 | 14.79 | 10 | 10 |
| Churu | 0.14 | 41 | 0.55 | 40 | −12.81 | 38 | 40 |
| Dausa | 0.14 | 36 | 0.54 | 32 | −7.27 | 41 | 35 |
| Deluxe | 0.09 | 32 | 0.47 | 25 | 2.24 | 25 | 25 |
| Dhaulpur | 0.15 | 27 | 0.47 | 24 | 1.4 | 30 | 24 |
| Didwana | 0.1 | 38 | 0.54 | 33 | −11.25 | 39 | 37 |
| Dungarpur | 0.17 | 23 | 0.43 | 23 | 3.54 | 23 | 23 |
| Falna | 0.05 | 49 | 0.79 | 49 | −26.75 | 48 | 48 |
| Ganganagar | 0.25 | 11 | 0.29 | 11 | 9.36 | 19 | 13 |
| Hanumangarh | 0.3 | 2 | 0.16 | 4 | 21.46 | 7 | 4 |
| Hindaun | 0.2 | 22 | 0.4 | 21 | 9.42 | 21 | 22 |
| Jaipur | 0.13 | 6 | 0.27 | 8 | 17.77 | 9 | 7 |
| Jaisalmer | 0.01 | 50 | 0.91 | 51 | −40.74 | 52 | 51 |
| Jalore | 0.08 | 40 | 0.53 | 37 | −11.24 | 36 | 39 |
| Jhalawar | 0.18 | 15 | 0.31 | 15 | 8.19 | 22 | 17 |
| Jhunjhunu | 0.25 | 8 | 0.28 | 14 | 21.89 | 5 | 9 |
| Jodhpur | 0.15 | 12 | 0.27 | 10 | 13.69 | 12 | 11 |
| karauli | 0.02 | 52 | 0.96 | 52 | −33.05 | 49 | 51 |
| Khetri | 0.08 | 47 | 0.69 | 47 | −16.85 | 42 | 46 |
| Kota | 0.14 | 20 | 0.33 | 18 | 7.68 | 20 | 19 |
| Kotputli | 0.21 | 29 | 0.52 | 31 | −1.87 | 31 | 31 |
| Lohagarh | 0.17 | 14 | 0.32 | 16 | 15.5 | 13 | 15 |
| Matsyanagar | 0.2 | 18 | 0.38 | 20 | 11.33 | 15 | 18 |
| Nagore | 0.1 | 30 | 0.47 | 27 | −6.6 | 32 | 30 |
| Pali | 0.08 | 44 | 0.68 | 45 | −18.43 | 43 | 43 |
| Phalaudi | 0.06 | 42 | 0.66 | 42 | −18.38 | 45 | 42 |
| Partapgarh | 0.04 | 51 | 0.91 | 50 | −40.07 | 51 | 50 |
| Rajasamand | 0.06 | 45 | 0.68 | 44 | −18.72 | 44 | 44 |
| Sardaarshahar | 0.14 | 24 | 0.5 | 30 | −4.41 | 34 | 29 |
| Sawaimodhopur | 0.06 | 48 | 0.81 | 48 | −31.55 | 50 | 48 |
| Shapur | 0.16 | 39 | 0.61 | 41 | −16.25 | 40 | 41 |
| **Sikar** | **0.78** | **1** | **0.05** | **1** | **30.34** | **3** | **1** |
| Sirohi | 0.08 | 43 | 0.68 | 43 | −17.04 | 47 | 44 |
| Srimadhopur | 0.22 | 19 | 0.38 | 22 | 9.42 | 17 | 19 |
| Tijara | 0.19 | 28 | 0.55 | 34 | 0.9 | 29 | 31 |
| Tonk | 0.16 | 21 | 0.36 | 19 | 10.6 | 18 | 19 |
| Udaipur | 0.13 | 9 | 0.21 | 9 | 21.37 | 4 | 6 |
| Vaishalinagar | 0.19 | 4 | 0.13 | 3 | 31.5 | 2 | 3 |
| Vidhyadharnagar | 0.23 | 34 | 0.49 | 35 | 3.24 | 26 | 34 |

## 5. Sensitivity Analysis

In general, the data obtained for MCDM problems are relatively imprecise and ambiguous. According to [50], even minor variations in relative weights may lead to significant variations in the overall ranking. Although such weights frequently depend entirely on subjective assessments, it is important to investigate the consistency of rankings across different criteria. The FAHP technique was utilized to derive the criteria and the category weights were examined in determining the dominance of each scenario. As perceptions of decision makers vary, the robustness of the obtained results was studied through sensitivity analysis. Initially, the weight value was modified by increasing or decreasing the criterion

weight by 5%, 10%, 20%, or 50%, respectively. When a criterion weight increases or decreases by 5%, 10%, 20%, or 50%, the remaining criteria must be proportionally adjusted to keep the criterion weight at 1. Considering this set of weights, the MCDM methods TOPSIS, VIKOR, and ELECTRE were used to examine each criterion.

This goal can be achieved through sensitivity analysis, which can be based on scenarios that represent potential future developments or various viewpoints on the relative relevance of the criteria.This analysis helps in checking the consistency of results, regarding whether the model or system works in the most favorable or unfavorable conditions. Sensitivity analysis was used in this paper to see how the ranking of depots varied as the weights of the criteria were changed. The sensitivity coefficient indicated that increasing or decreasing the criterion weight by 5%, 10%, 20%, or 50% resulted in single, double, or multiple changes in the rankings of alternatives. The sensitivity coefficient was equal to 0 if the rank was the same as the original rank. When the rank of one depot increased, the rank of another fell, resulting in a sensitivity coefficient of 2.

The number of criteria that affected the rankings after adjusting the weight of one criterion is displayed in Table 8. The results demonstrated that, when weights were increased (decreased) by 5%, the ranking of depots had some impact on the ranking with the VIKOR and ELECTRE techniques but no impact on TOPSIS. When the criterion weights were increased or decreased by 50%, the ranking of the TOPSIS technique was the least affected, whereas ELECTRE showed the most significant change (46% and 69% change) and VIKOR exhibited a change (39% and 72% change). Only the TOPSIS ranking results remained nearly unchanged when the weight was varied drastically (50%), whereas the remaining two methods were altered by roughly 39% to 72%. In VIKOR, the top-valued weights were much more affected when weights were decreased, compared to when they were increased. For example, when the weight of the operated vehicle (C11) criterion was increased by 10%, we observed 21 ranking changes, but when it was decreased by 10% we observed almost twice as many ranking changes, i.e., 39. On the other hand, in the ELECTRE method, the operated vehicle (C11) criterion ranking was not affected, even though it was affected in all other models. It was expected that the ranking changes would have been high, but this was not the case for ELECTRE. "Operational income per km" (C43) was the least affected criterion in all the models. Among the considered scenarios, we observed extreme deviations in punctuality (C21), vehicle utilization (C23), fleet utilization (C22), rate of breakdown (C24), passenger km occupied (C32), number of passengers (C31), and load factor (C32).

**Table 8.** Sensitivity analysis of criterion weights.

| Change in Criterion Weights | TOPSIS | | | VIKOR | | | ELECTRE | | |
|---|---|---|---|---|---|---|---|---|---|
| | 0 | 2 | >2 | 0 | 2 | >2 | 0 | 2 | >2 |
| 0.05 | 10 | 2 | 3 | 0 | 5 | 10 | 3 | 2 | 10 |
| −0.05 | 8 | 2 | 5 | 0 | 6 | 9 | 0 | 2 | 13 |
| 0.1 | 7 | 2 | 6 | 0 | 0 | 15 | 2 | 2 | 11 |
| −0.1 | 8 | 1 | 6 | 0 | 2 | 13 | 0 | 0 | 15 |
| 0.2 | 7 | 1 | 6 | 0 | 0 | 15 | 1 | 0 | 14 |
| −0.2 | 7 | 1 | 6 | 0 | 0 | 15 | 0 | 0 | 15 |
| 0.5 | 6 | 2 | 7 | 0 | 0 | 15 | 0 | 0 | 15 |
| −0.5 | 6 | 3 | 6 | 0 | 0 | 15 | 0 | 0 | 15 |

## 6. Conclusions

A crucial factor in the development of a nation's economy is its public transportation sector. The transport sector is difficult to fix and includes a number of systems related to the categories of operational service, service quality, passenger service, and cost implications. Reduced cost effects and the provision of optimum quality byan inefficient depot must be provided through the depot's efficient operation. In the transportation sector, the correct selection of criteria is crucial to avoid the degradation of service quality at bus depots. To assess the most appropriate criteria for the optimum functioning of depots, in this work we incorporated a novel hybrid MCDM method that combined FDM, FAHP, TOPSIS, VIKOR,

and ELECTRE. The critical criteria were computed using FDM and the weights of the assessment categories and criteria were measured using FAHP. When evaluating the depots based on performance scores, TOPSIS, VIKOR, and ELECTRE received the FAHP weights as their input. The MCDM techniques yielded noteworthy findings, bridging the gap between earlier research in the transport sector and the challenge of criteria selection. In this study we investigated the outcomes of the suggested hybrid paradigm. The findings obtained using the proposed method have been presented in tabular form. Different evaluation criteria were used in earlier studies, including total vehicles, scheduled vehicles, operated vehicles, off-road vehicles, and many more. However, in this study we assessed a few significant factors in the FDM performance evaluation procedure. The suggested approach is a straightforward, convenient, precise, and effective instrument that can assist decision-makers with the selection of criteria for the performance management of depots. These novel hybrid MCDM methods can handle various criterion selection problems in the transportation sector and can be applied in decision-making contexts as well. As a result, we made an effort to obtain the significant criteria for depots by utilizing a variety of methods. Future scholars could further investigate challenges related to the transport sector and researchers interested in the topic of criteria selection using MCDM approaches may find this article beneficial.

This study also has some limitations. The proposed methodology can be further developed in the future in order to identify and select the most suitable criteria, which is very challenging in relation to implementing sustainable policies in the public transport sector, using DEMATEL, ANP, etc. Furthermore, by using cross-sectional data in case studies, this work can be extended with an approach that will provide a more accurate ranking of bus depots.

**Author Contributions:** Conceptualization, S.G.; Data curation, S.G.; Formal analysis, S.G.; Investigation, S.G.; Methodology, S.G.; Project administration, S.A., T.M.; Resources, S.G.; Software, S.G., S.A.; Supervision, S.A., N.S.S.S., T.M. and N.M.; Validation, S.G., S.A., N.S.S.S., T.M. and N.M.; Visualization, S.G., S.A.; Writing—original draft, S.G.; Writing—review & editing, S.G., S.A. All authors have read and agreed to the published version of the manuscript.

**Funding:** This research received no external funding.

**Institutional Review Board Statement:** Not applicable.

**Informed Consent Statement:** Not applicable.

**Data Availability Statement:** Not applicable.

**Acknowledgments:** The authors are thankful to Harish Puppala for his valuable support. The authors are also thankful to the Faculty of Data Science and Information Technology, INTI International University, Malaysia, for guiding and supporting them in providing resources for this research.

**Conflicts of Interest:** The authors declare no conflict of interest.

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
