# Peer review of "Analysis of Hybrid MCDM Methods for the Performance Assessment and Ranking Public Transport Sector: A Case Study"

_sustainability, doi:10.3390/su142215110_

Round 1

Reviewer 1 Report

Dear authors

I believe that your work needs a thorough rewriting and most fundamental, using the correct MCDM tools for this problem.

As I said in my comments, AHP is not the right method to use because the problem contains some characteristics that this method can’t handle, such us: Relationship between criteria and no hierarchical linearity.

But what surprised me the most, was the mix that you apply in determining bus depots performance and using about 25 criteria that have no relation with them. As I can see it, you are trying to solve two different problems, performance of bus depots, and operation performance and the same time. Well, at least, this was my feeling when reading it.

Please, don’t take offense, but I would recommend to consult somebody with a good working knowledge of this kind of operation.

There is something that really puzzled me.

 Maybe 10 years ago I boarded the sleeper bus at Jaisalmer ( in a square) to Agra as a final destination, which I believe is a very important route in Rajasthan. I clearly remember that the bus stopped at road side markets, without any resemble to a bus depot, and of course, with nothing for the comfort of travellers, let alone with a bath room. Maybe it has changed, and now they are stations or bus depots, therefore along the reading of your paper I was questioning myself ‘Where are the bus depots that  this people are talking about?

Author Response

Please see the attachment. Kindly, let me know if any improvement is needed.

Reviewer 2 Report

The subject is interesting, however, several points have to be improved.

-Keywords: What is FAHP, you should clarify it as only AHP, since you already  used Fuzzy set theory.

-Saaty said there is no need to use fuzzy sets with AHP, because AHP is already fuzzified, please, justify using fuzzy sets with AHP.

-Justify why you have selected TFN not other type.

-The introduction part is limited and need to be extended, there are several works discuss using AHP with other applications and fuzzy sets to evaluate public transport system, you have to discuss them in the introduction, I recommended the following related works:

-- Sustainable urban transport planning considering different stakeholder groups by an interval-AHP decision support model

--Evaluating public transport service quality using picture fuzzy analytic hierarchy process and linear assignment model

--Examining Pareto optimality in analytic hierarchy process on real Data: An application in public transport service development

--Analysing stakeholder consensus for a sustainable transport development decision by the fuzzy AHP and interval AHP

--An integrated grey AHP-MOORA model for ameliorating public transport service quality

Figure 2, where is AHP?

4.2. Fuzzy Analytic Hierarchy Process

Highlight the model with a numerical example, with calculation process.

Same examples have to be provided for all used methods in the paper.

_ You have to use journal's style.

Author Response

(The authors gave the same response as above.)

Reviewer 3 Report

The manuscript must be seriously modified to be acceptable. Please improve the quality of the paper using the following remarks.

The manuscript has not been prepared following the journal's requests. The official template isn't used, the cite style isn't appropriate, etc.

All abbreviations must be defined properly. For example in abstract: TOPSIS, VIKOR, ELECTRE.

Abstract must be rewritten. Please write it in scientific way containing all the important elements: goals, novelty, contributions.

Introduction must be extended, missing motivation, exact goals, contributions...

The structure of the paper should be moved in introduction.

Figure 1 must be cited above Figure, not below.

The literature review should be updated with more wider field related to transport. Please cite the following studies:

Sénquiz-Díaz, C. (2021). Transport infrastructure quality and logistics performance in exports. ECONOMICS-Innovative and Economic Research, 9(1), 107-124.

Sénquiz-Díaz, C. (2021). The Effect of Transport and Logistics on Trade Facilitation and Trade: A PLS-SEM Approach. ECONOMICS-Innovative and Economic Research, 9(2), 11-34.

Please in concise way explain why you using Fuzzy AHP. This method is old and with some disadvantages.

- You should obtain statistical tests of correlation. 

- Add limitations and implications.

Author Response

(The authors gave the same response as above.)

Round 2

Reviewer 2 Report

The authors improved the paper accordingly.

Author Response

Thank you for your comments. Your comments have helped to enhance the quality of the article. 

Reviewer 3 Report

The paper can be published in modified version.

Author Response

(The authors gave the same response as above.)
